# Production of a Rich Fertilizer Base for Plants from Waste Organic Residues by Microbial Formulation Technology

**DOI:** 10.3390/microorganisms12030541

**Published:** 2024-03-07

**Authors:** Sai Shiva Krishna Prasad Vurukonda, Vasileios Fotopoulos, Agnieszka Saeid

**Affiliations:** 1Department of Engineering and Technology of Chemical Processes, Faculty of Chemistry, Wroclaw University of Science and Technology, 50-370 Wrocław, Poland; 2Department of Agricultural Sciences, Biotechnology & Food Science, Cyprus University of Technology, Limassol 3036, Cyprus; vassilis.fotopoulos@cut.ac.cy

**Keywords:** sustainable agriculture, bioformulation, nutrient loop, waste processing, soil microorganisms

## Abstract

This review explores different methods of sustainably introducing nutrients from agro-industrial waste into the soil. The focus is on sustainable agriculture and how the soil system can be modified by introducing secondary raw materials and beneficial microorganisms. Soil is a nexus between plants and microorganisms that must be balanced. The article emphasizes the importance of maintaining the microbiological balance when supplying nutrients. This review is focused on the possible techniques involved in the production of biofertilizers and their mode of application into the soil system and on plants. We addressed several advantages concerning the use of beneficial microorganisms in waste management by microbial formulation techniques. Not only the advantages but several limitations and challenges were also discussed in regard to the large scale production of microbial products. Therefore, the proper treatment of industrial waste is essential so that we can preserve the environment and human safety and also achieve sustainable agriculture.

## 1. Introduction

Considering the exploitation of resources (water, land, energy) related to agricultural activities, the valorization of agro-food wastes and by-products is necessary from both an environmental and socio-economic viewpoint. Therefore, many attempts are made nowadays to develop nutrient (N, P, K, and microelements) recovery techniques from waste streams to reduce the environmental impacts linked to their dispersion [1,2]. The currently developed waste treatment methods mainly consider thermal and chemical processes focused on removing the organic fraction and ensuring biological safety [3]. The equilibrium of the soil and rhizosphere’s microbiota could be significantly affected, and nutrient-rich secondary raw materials can offer an improved mineral nutrition, if the new formulations avoid overloading the soil-plant nexus with potentially toxic substances [4].

The microbiological stability of the soil is another important issue within the sustainability of agriculture, often neglected when discussing nutrient recycling. Soil is the very rich, complex link connecting two different subsystems. It creates an environment for the growth of two groups of organisms: saprotrophs and producers. Saprotrophs are responsible for fostering the releasing nutrients from dead organic matter into the form available to plants [5]. That is why soil fertility relies not only on a balanced level of nutrients but also on the richness of the soil biota [6]. Thus, the saprotroph subsystem consists mainly of microorganisms, and is responsible for re-introducing nutrients into cycling throughout the ecosystem. However, the function of this link is not limited only to closing the loop of nutrients. Saprotrophs are also responsible for stabilizing the unwanted toxic elements present in the soil via bioaugmentation, biosorption, bioaccumulation, and bioremediation processes [7]. That is why its biodiversity and abundance should be of special concern right next to nutrient recirculation.

The strategy of the simultaneous use of secondary raw material-based fertilizer products with beneficial microorganisms as stabilizers of soil microbiota has not been extensively discussed or considered so far. However, such integration of the application of the source of nutrients with microorganisms could enhance their cycle by boosting the stability of the soil microbiota. One of the few valorization methods of nutrients into fertilizer products that engage the microbial activators is bio-solubilization [8]. Through microbial metabolic activity, compounds present in the starting materials are modified, and the substrates undergo a profound change, leading to the final products with different characteristics. Furthermore, the products of metabolism, such as amino acids, organic acids, and others, effectively liberate nutrients making them available for plants.

To ensure the stability of the agricultural system, it is crucial to provide microbial activators that will increase the availability of nutrients from secondary raw materials. In addition, they can positively change plant growth capacities and make them more resistant to biotic and abiotic stresses than applying fertilizer products based on secondary raw materials [7,9]. Still, very little is known about the relationships that govern the availability of nutrients in modified soil systems. There is, therefore, a need to increase the knowledge about how new soil systems would work where secondary raw material-based products would be used as a source of nutrients simultaneously with beneficial microbes. In addition, there is a need to elaborate the method for the efficient introduction of microbial activators to the soil system and understanding how it will influence the release of nutrients and the equilibrium of the soil microbiota and soil status.

To date, only a few methods of the microbial valorization of ‘waste’ organic matter for fertilizer preparation have been described. These methods have been used mostly to determine the total available phosphorus content or to assay the biological response activity of the used microorganisms in sterile rhizosphere soil. Some of this research showed the biological activity of amino acids, organic acids, and other extractable components. However, the absence of thorough studies on non-sterile rhizosphere soil inside the ‘waste’ organic matter valorization strategy restricts our capacity to identify correlations between microbial community structure and rhizosphere chemistry [10]. Edaphic and environmental circumstances are two significant factors influencing biofertilizer variability and low repeatability in field experiments. The initial stages of biofertilizer testing are carried out in aseptic circumstances, which do not allow for an unbiased characterization of the microorganism under investigation. However, scaling up the growth chamber or greenhouse trials, and especially to field settings, expands the range of uncontrolled biotic and abiotic elements that can have a major impact on performance. The most often cited biotic elements that might influence the outcome include competitors, predators, or other antagonists within the native microbiome (i.e., indigenous and previously imported bacteria). Abiotic conditions, whether climatic or edaphic, can also impact the effectiveness of bio-fertilization on crop nutrient use efficiency and output [4,5].

The bacterial extraction of nutrients from waste streams is currently attracting a lot of attention due to its sustainability [11,12]. The microbial solubilization process requires an interaction between the microbes and the substrates. Thus, selecting appropriate strains is crucial to guarantee the adaptation of microorganisms to the process and complete bioconversion of the starting materials. Microbially assisted solubilization processed waste streams can improve microbial stability by supplying specific nutrients and supporting an ecological niche for inoculated microorganisms. The research scope will assess the efficacy of different types of groups of microorganisms capable of digesting waste matter, the mechanism of its digestion/solubilization, the effectiveness of the different methods of infestation of soil system, and their efficiency in real conditions via a field test.

## 2. Complexity of the Soil System

The soil system is a complex system of interactions among the three pillars that influence its qualities as mentioned in Figure 1.

The fertility of the soil in the rhizosphere is influenced by plant roots and a group of soil microorganisms. This eventually affects the growth, yield, composition, and nutritional qualities of the plant biomass that is produced. Each plant shapes the environment around it in a specific way by influencing its chemical, physical, and biological properties. Plants stimulate various soil flora and fauna groups through secretions and post-harvest residues of different chemical compositions. Microorganism metabolism by-products act as biostimulants, stimulating the release of nutrients present in the soil in an inaccessible form which the plant root system can take up. In turn, plant roots interact with microorganisms, improving growth, nutrient acquisition, and protection from different plant bio-aggressors. At the same time, it was found that a low level of nutrients in the rhizosphere can stimulate root tissues to release exudates that can stimulate the growth of microorganisms. That, in turn, increases the availability of missing nutrients [13].

Plant roots exude around 11–40% of their photosynthetically fixed carbon into the soil, known as root exudates. Root exudates and mucilage serve as nutrition supplies and signaling molecules for soil microorganisms, altering the microbial population around the root system. Root exudates could affect rhizosphere interactions through selective biocidal and/or signaling activities, in addition to being the main source of carbon for rhizosphere microbes. Rhizosphere interactions have been observed to be influenced by both polar and non-polar substances. More complex non-polar secondary metabolites, including flavonoids, coumarins, and benzoxazinoids, have been shown to have a significant impact on rhizospheric bacteria, in addition to polar primary metabolites like organic and amino acids. Microbial mechanisms, on the other hand, assist plant development by supporting nutrient uptake, plant growth hormone synthesis, and pathogen biocontrol [12] (Figure 2).

Interactions between roots and the microbial community also impact the physicochemical properties of the surrounding soil. Spatial and temporal dynamics influence the interaction of the various rhizospheric components, resulting in dynamic feedback loops that preserve the complex rhizosphere environment with physical, chemical, and biological gradients that are different from the bulk soil. Gaining an understanding of these complex rhizospheric connections is essential to understanding localized biogeochemical processes and developing ways to boost plant yield [14].

These complex relationships are controlled by the currently prevailing physicochemical conditions in the soil, the type of plant and the biodiversity of the soil flora. Each of the pillars mentioned above was and is often considered independently of the others. Therefore, the soil system should be understood as a set of these three to achieve a balance in the soil system and fully understand the binding relationships. The soil fauna is the most frequently overlooked in the context of sustainable development in agriculture. However, it is recognized how soil movement contributes to soil fertility when attempting to explain how microbial activity might promote plant development, three methods are typically mentioned: (i) modifying plant hormone signaling; (ii) preventing or outcompeting pathogenic microbial strains; and (iii) boosting soil-borne nutrient bioavailability (Figure 3).

### 2.1. Bio-Waste Materials as Source of Nutrients

Biofertilizers are widely used in organic farming systems. However, there is still little known about the mechanisms of the process, especially in the real agricultural environment. For example, how will the choice of plant cultivars, type of ‘wastes’, microbial inoculants, and the method of stable introduction of biofertilizers into the soil system influence the effectiveness of the process? All the proposed waste compounds (Table 1) can be subjected to microbiological treatment using biofertilizers. As a result, it is possible to release nutrients/biostimulants from the waste matter matrix. The individual cases have been well described in the literature, indicating the microorganisms and their effectiveness. However, in most cases, these are descriptions of the results of laboratory tests using single strains and one type of waste [8,15]. In this context, detailed research is necessary, including screening of microorganism systems that will effectively dissolve waste materials in natural systems, including variations in the method of introducing microorganisms.

For instance, research on keratinases and the in vitro degradation of feathers revealed that keratin cannot be broken down by a single keratinase enzyme. The activity of keratinases is insufficient to break disulfide bonds. The following processes—sulfitolysis, proteolysis, and deamination—have been identified as potential contributors to keratin degradation, and many methods have been put forth. To comprehend the mechanism of action of keratinases, more research on keratin degradation is necessary [23]. Using many strains together might be an effective way to break down feathers. In the area of microbial degradation, it has been observed that microbial consortia, or combined microorganisms, have distinct impacts than a single microbe. A comparable method may be applied to choose appropriate stains that effectively break down waste biomass [23].

### 2.2. Nutrient-Rich Formulations of Biowaste Materials

Over 120 million tons of bio-waste are thought to be produced in Europe each year. A tiny portion of bio-waste is burned without recovering the abundance of valuable nutrients, and the remainder is landfilled [24]. The amount of municipal solid trash generated worldwide is estimated to be 1.3 billion Mg/year, and it is anticipated to grow in the coming years [25]. A significant portion of municipal waste, which includes food waste and green & garden waste (leaves, grass), is organic waste. There are several scenarios that may be used to handle it, depending on its composition and humidity [25]. The treatment of biowaste can help with the synthesis of new molecules, as well as the recovery of resources and energy. Additionally, these residues include a wealth of valuable substances, such as phenolic compounds [26], vitamins, carotenoids [27], as well as proteins [28,29].

The current necessity for enhancing crop productivity, soil fertility, and pest control in sustainable farming is the use of bio-based supplies to reduce the application of chemical-based products and their related detrimental effects on the environment. Wasted coffee grounds, wasted mushroom substrate, paddy straw, blood meal, biochar, and other significant biowaste substances are recognized to be essential for plant metabolic pathways, enhancing microbial interaction, and promoting host resistance while inhibiting pathogen colonization. These characteristics support the potential use of nutrient-rich compounds as efficient biopesticides and fertilizers [30,31].

The term “soil biofertilizer” refers to microorganisms that enrich soil with nutrients and carbon substrates, including bacteria, fungi, algae, and cyanobacteria. The most common and extensively utilized biofertilizers are green manures, which include cyanobacterial supplements and bioformulations of bacteria like *Azotobacter* sp., *Azospirillum* sp., *Trichoderma* sp., and arbuscular mycorrhizal fungus (AMF). Farmers commonly use organic-based fertilizers, such as crop residues, vermicomposting residues, farmyard manure, and other waste substrates, in addition to microbial biofertilizers [32,33]. Solid-state fermentation was used to create a biofertilizer, which was subsequently applied to a vegetable garden. The physical characteristics of plant samples treated with biofertilizers were positively represented by the testing results. The two main procedures that utilize the metabolic potential of the thermophilic and decomposer bacteria populations are composting and anaerobic digestion (AD) [34]. Native microbial communities have enzymes that support bioprocesses that turn agricultural waste into biofertilizers [35].

### 2.3. Microorganisms

Implementing the assumptions of sustainable agriculture in the European Union required greater attention to the beneficial effects of using biostimulants, which, by definition, do not provide nutrients themselves but allow for achieving better crop parameters with significantly reduced fertilization with chemical fertilizers. Although the term ‘biostimulant’ has become a permanent feature of scientific and professional literature, it was first used only in 2007 [36]. Earlier, terms such as materials that, in minute quantities, promote plant growth, ‘hormone-containing products’, or metabolic enhancers were used. Nevertheless, throughout the years, the term “biostimulant” has been used more frequently in scientific publications, encompassing a wider variety of compounds and mechanisms of action. Currently, biostimulants are classified into five classes: (i) seaweeds and plant extracts; (ii) humic substances; (iii) hydrolyzed proteins and nitrogen-containing compounds; (iv) microorganisms; (v) inorganic compounds with biostimulant action [37].

The increase in their importance in agricultural practice is largely based on their very favorable properties, which concern not only the crop quality and improved performance under stress conditions [38,39], but also the reduction of the negative load on the natural environment, achieved by reducing the need to use artificial fertilizers, contributed to the change in the fertilization law. Biostimulants are included in the new Regulation (EU) 2019/1009 of the European Parliament and of the Council, which was published on 5 June 2019. It also makes a distinction between microorganisms and their exclusion from the regulations. Presently, the following goods fall under the purview of the relevant EU regulation when it comes to fertilizer products: (i) fertilizer; (ii) liming material; (iii) soil improver; (iv) growing medium; (v) inhibitor; (vi) plant biostimulant; (a) microbial plant biostimulant; (b) non-microbial plant biostimulant; (vii) fertilizing product blend.

According to the aforementioned EU legislation, a plant biostimulant is any substance that enhances the plant’s feeding process regardless of its nutritional content, with the express purpose of enhancing one or more of the following traits of the plant or its rhizosphere: (i) efficiency in using nutrients; (ii) ability to withstand abiotic stress; (iii) characteristics of high quality; (iv) accessibility of restricted nutrients in soil or rhizosphere. In other words, it is the group of preparations that improve plant growth but do not provide nutrients themselves. Biostimulants support the natural processes in plant tissue due to the presence of such substances as beneficial soil microorganisms, and phytohormones such as auxins, cytokinins, or amino acids. In this class of compounds/substances, we distinguish a group of products referred to as bio-fertilizers.

Microbial biofertilizers are made up of beneficial microorganism cells that interact with the rhizosphere or endosphere of plants and have the ability to promote plant development. They use elements already found in the soil to supply materials that promote plant development. By encouraging nutrient absorption and ultimately increasing productivity, they enhance soil fertility [8,40]. There are many commercial products formulated using selected soil microorganisms, whose main purpose is to increase the availability of nutrients. In addition, some preparations contain growth biostimulants that increase biotic and abiotic stress resistance, although they do not provide nutrients themselves [7,9].

## 3. Overview of Different Forms of Microbial Formulations

### 3.1. Microbial Formulation Technology

A significant obstacle in soil ecology is the diverse and dynamic soil microbial population, which varies in composition between various compartments and layers. The possible effects of inoculation on the environment were never taken into consideration. Inoculation would cause at least a temporary disturbance of the balance of soil microbial communities since it provides large densities of effective and viable microorganisms for fast colonization of the host rhizosphere. If significant native species disappear, it might negatively impact future harvests by altering the microbiota and causing unfavorable changes. However, the degree of variety and interactions among the plant, soil, and microbiota may function as a buffer against inoculation-induced alterations in the microbe population structure [41]. Because several bacterial species may perform the same tasks, bacterial redundancy may prevent the loss of certain species from impairing the system’s ability to function properly [42]. Microbial formulations are a viable substitute for chemical inputs as microbial inoculants are ecologically benign, or “eco-friendly”. They could be microbial biocontrol agents, biofertilizers, or phytostimulants. Bioherbicides can also be made from naturally occurring bacteria that have been separated from their natural environment and sprayed on plants. Relatively large quantities of microbial cells are added to the competitive soil environment in order to employ microbial inoculants [43]. A variety of microorganisms have been investigated and are often employed as microbial antagonists. The discovery of new species, the selection and enhancement of established strains, and the introduction of non-native genes to obtain expressed products or novel functional features have all contributed to recent advancements in classical microbiology. We refer to this intricate and useful branch of microbiology as technological advances in microbiology [44]. Different organic and inorganic carriers are used in the formation of the prospective microbial isolates utilizing either liquid or solid formulation techniques. They are applied by soil and foliar application, seed treatment, biopriming, seedling dip, or combinations of strains as co-inoculants or consortiums [45].

#### 3.1.1. Single Inoculants

Nowadays, a wide variety of single microorganisms are sold as microbial inoculants. Several fungi pathogenic to insects are also being used as control agents, including *Beauveria*, *Metarhizium*, *Verticillium,* and *Paecilomyces*. These are most commonly used in greenhouses and other locations with reasonably high humidity to combat leaf caterpillars. Numerous arthropod species are susceptible to *Beauveria bassiana*’s effects. When employing preparations of fungal spores, the effectiveness of fungal microbiological therapies is significantly influenced by environmental parameters, including temperature and humidity. Establishing the infection, however, may cause insects to die long before chemical controls take effect. Both the impacts on non-target species and the likelihood of resistance developing are significantly lower. Spores often adhere, germinate, and penetrate insect cuticles when they come into contact with them. Naturally, a large range of toxins and molecules that cause behavioral changes or modifications are known to be released by several entomopathogenic fungi (*Cordycipitales*, *Trichocomaceae*, etc.). Several *Lepidopteran* insect larvae have been effectively controlled by the fungus [46]. C. F. Von Tubeuf was the first to create the phrase “biological control” as a viable aspect of plant disease management in 1914. Since then, it has been discovered that a variety of biocontrol agents are highly successful at managing plant diseases. Sanford [47] discovered that green manuring antagonistic activities inhibited potato scab. Weindling [48] showed that *Trichoderma lignorum* is a parasite on a number of plant diseases. Grossbard [49], Wright [50] and others showed that *Penicillium*, *Aspergillus*, *Trichoderma*, and *Streptomyces* species generated antibiotics in soil. Kloepper [51] showed how important the siderophores produced by *Erwinia carotovora* are. Howell [52] reported P and Q strains of *Trichoderma* sp. [53]. A suspension of *Phytophthora palmivora*’s chlamydospores was the first commercially registered mycoherbicide, used to control *Morrenia odorata* [54].

Several microorganisms such as *Trichoderma harzianum* [55] *Pseudomonas fluorescens* [56], and *Bacillus subtilis* [57] can control many foliar and soil-borne fungi, i.e., *Fusarium* spp., *Rhizoctonia solani*, *Pythium* spp., *Sclerotium rolfsii* in vegetables, fruits, and industrial crops [58]. The goal of *Trichoderma* is to produce mycoparasitic strains that are efficient in controlling plant fungal diseases in a variety of environmental settings [59]. It was discovered that some *Bacillus* species penetrate the root surface, promote plant development, and induce fungal mycelia to lyse [60,61]. Because, *B. subtilis* cells can produce dormant spores that withstand harsh environments, they are simple to prepare and store [62]. Additionally, *B. subtilis* generates a wide range of physiologically active substances that exhibit a broad range of actions against phytopathogens and have the ability to cause systemic resistance in the host [63]. It has also been demonstrated that a number of *B. subtilis* strains are able to create multicellular structures, or biofilms [63]. These advantageous characteristics make *B. subtilis* a possible candidate for use as a biological control agent. According to reports, certain strains of *B. subtilis* can successfully control the wilt disease caused by *Ralstonia* spp. in a variety of plant hosts. [64,65,66]. In agricultural environments, bacteria, mycorrhiza, and other fungi may all help to promote growth while providing biocontrol. These organisms may function as biocontrol agents, minimizing pathogenic agent damage to plants, or they may modify the levels of critical plant hormones such as auxin and ethylene. They may also assist the plant in acquiring necessary resources like iron, phosphate, nitrogen, or water [67]. The genera *Rhizobium*, *Sinorhizobium*, *Mesorhizobium*, *Bradyrhizobium*, *Azorhizobium*, and *Allorhizobium* include the most effective N-fixing strains [68]. *Pochonia chlamydosporia* [69,70], and *P. fluorescens* [71] can effectively control diseases caused by nematodes [72]. Significant advancements have been made in the development and distribution of bionematicides in recent years [73]. Avermectin chemicals, which are the secondary metabolites of the bacterium *Streptomyces avermitilis*, serve as model pesticides because they are effective against nematodes even at extremely low dosages and are not harmful to mammals. Therefore, adult nematodes and larvae, such as *Radopholus similis*, *Meloidogyne incognita*, and *Ditylenchus dipsaci*, become immobile and die when exposed to filtrates of *Bacillus firmus* cultures. This indicates that the production of hazardous compounds has a role in the management of these pests [74].

#### 3.1.2. Co-Inoculants

The reported variations in the field may be somewhat explained by the fact that most inoculants frequently rely on administering a single strain. Adding many beneficial microbial species or strains to the same microbial mixture is one method to overcome this issue. Without requiring genetic engineering, co-inoculation combines many ways to improve plant performance and the effectiveness and dependability of beneficial impacts on crops [75]. In order to improve plant production and physiological parameters and to regulate plant health, co-inoculation of biocontrol agents and biophytostimulators is considered to be a beneficial strategy. By utilizing a variety of modes of action, the application of distinct microbes can increase the range of biocontrol activity, improve the efficacy and dependability in suppressing disease incidence, and promote plant growth without the need for the use of genetic engineering techniques [76]. Soil contains an indefinite number of microbes, including deleterious organisms. The isolation and evaluation of several beneficial microbes with various modes of action concerning biological control and plant growth promotion is an emerging area of research. The introduction of beneficial microbes isolated from the soil ecosystem closely represents the natural situation and can replace deleterious microbes when applied at higher inoculum levels. Moreover, applying mixtures of beneficial microbial cultures enhances the efficacy and reliability of plant health management [77]. However, certain mixtures of microbial strains negatively affect the suppression of pathogen infection and plant growth promotion [78]. Additionally, co-inoculants made up of many AM (*Arbuscular Mycorrhiza*) fungi can be created. For instance, it has been demonstrated that the AM fungus *Funneliformis mosseae* can systematically lower the disease infection in barley caused by *Gaeumannomyces graminis* var. *tritici* [79]. Thygesen et al. [80] discovered a potential mycorrhiza-induced resistance against the pathogen *Aphanomyces euteiches’* ability to cause pear root rot. In addition, differences in the level of tolerance induction were observed between the two AM fungi that were employed, *Glomus claroideum* and *G. intraradices*. Abdel-Fattah et al. [81] reported that in both greenhouse and field experiments, treatment with a combination of AM fungi (*Glomus intraradices*, *G. mosseae*, *G. clarum*, *Gigaspora margarita*, and *G. gigantea*) effectively decreased the white root disease in onions caused by *Sclerotium cepivorum*.

#### 3.1.3. Microbial Consortia

Since bacteria do not exist in isolation in the natural environment, a group of bacteria may be more beneficial for promoting plant development than a single bacterium [82,83]. The development of bacterial consortia and their composition, however, is a difficult task as the members of the bacterial mixture must be compatible with one another. The bacteria selected for the bacterial consortium should also be adaptable to the unfavorable circumstances present in agricultural fields and have a variety of capacities to promote plant development and perform bioremediation. There is a greater likelihood that one bacterium in the bacterial combination will carry out the functional gene expression needed for plant growth promotion, which is another significant reason why bacterial consortiums may operate better than single-bacterium formulations [84].

Several studies have demonstrated that a single strain cannot completely degrade contaminants. As various strains have diverse metabolic pathways, bacteria that have various reduction capabilities are combined, and the microbial consortium may incorporate each strain’s advantages to ensure effective pollutant degradation [85]. Mixed microbial consortia demonstrated high performance in substrate tolerance and enhanced pollutant breakdown [86,87,88,89,90,91,92]. The consortia of microorganisms outperform single-strain cultures. The microbial consortium demonstrated apparent impacts in the degradation of contaminants [86]. Some relevant microbial strains isolated from the intestinal flora and natural flora have the intrinsic potential to breakdown contaminants [86,93]. *Lactobacilli*, *Actinobacteria*, *Pseudomonas*, *Clostridium*, *Salmonella*, and *E. coli* have been found to have the inherent ability to degrade pollutants [86]. These strains are suitable for the bioremediation of pollutants [86,94]. The microbial consortium has developed into a vital tool since it reduces contaminants more effectively than a single strain [86,93]. Bioremediation is often carried out by a microbial consortium rather than individual species in the natural environment, and various strains or species perform distinct functional tasks. Co-cultivating the microbial consortium is more effective than single bacteria, destroys pollutants more quickly, and may considerably boost the bio-degradation of contaminants in the soil [91].

Considering both economic and environmental perspectives, sterilizing biomass wastes is impracticable. Therefore, research in the applied sciences of synthetic microbial consortia (SMC) should focus on non-sterile conditions. However, the native microbial structure in biomass wastes is complex and variable, particularly in continuous or semi-continuous processing where continuous substrate addition significantly influences the existing microbial community structure. This implies higher requirements for the robustness and resistance of SMC to perturbations. Immobilization methods may reduce or eradicate this problem. Moreover, commercialization, which provides notable benefits in preserving microbial activity during the storage and transportation of SMC, depends on immobilization as well [95]. In Table 2 shows some examples of successful applications of microbial consortia in the degradation of various waste materials and the release of useful compounds.

### 3.2. Delivery Methods of Microbial Inoculants

Microbial inoculants can be applied to plants through various methods depending on the pathogen’s survival and mode of infection. These methods include seed treatment, root dip, soil application, and irrigation water. Effective formulations should ensure that the biocontrol agents are delivered in their active state at the right time and place. The microbial products should also adhere to plant components such as seeds, tubers, cuttings, seedlings, trans-plants, and mature plants, and they should be simple to apply and consistent with agronomic procedures. Even under adverse biotic and abiotic stress conditions, the microbial products should be available in the soil for an extended period of time.

#### 3.2.1. Seed Treatment

The delivery of the agents to the spermosphere of plants, where generally very favorable conditions predominate, will be substantially aided by the biofortification of seeds. Thus, the BCAs are given a great chance to endure, proliferate, and exert control on soilborne bacterial and fungal phytopathogens [103]. Microbial agents may be delivered “in the right amount, at the right place, and at the right time” with the use of precise treatment methods. The use of microbial inoculants in seeds is anticipated to rise in the future due to growing public awareness of the possible risks of agrochemicals pose to the environment and human health as well as the advancements in biotechnology that can enhance the performance of microbial products [104]. End users are continuing to pay special attention to this approach, which aims to deliver the active substances as near to the objective as feasible. Consistent work performed globally has led to significant advancements in seed treatment technology. This approach is appealing for introducing biological control agents into the soil–plant environment because it provides the introduced organisms the selective advantage of being leading colonizers of plant roots and increasing their number in the rhizosphere. The prepared product (liquids, powders) may be applied directly to seedlings without the need for adhesives. In order to promote product adherence to seeds, powders for seed treatment are created by combining an active agent with an inert carrier [105]. Additives like xanthan gum and gum arabic are frequently utilized to boost the microbial agents rate of survival when applied to seeds. When employed as a seed encapsulation medium, alginate hydrogel keeps the entity alive and shields it from several types of stress. Through a process known as “seed priming”, which involves combining seeds with an organic carrier and bringing their moisture content down to a level slightly lower than necessary for seed treatment—a technique that has been utilized to provide *T. harzianum* to reduce *Pythium*-induced damping off on cucumbers [106]. Another method of treating seeds involves applying *Trichoderma* spp. On radish and cucumber seeds through an industrial film-coating process that was developed for the application of chemicals and biological crop protection agents. It has been demonstrated to be effective against damping-off [107].

#### 3.2.2. Biopriming of Seeds

Biopriming is the hydration of seeds utilizing a biological or a biocontrol ingredient. Due to the discovery of seed priming technology, extensive research has been conducted on this type of seed treatment, which is now widely used for postponed planting and to promote faster plant development. As defined by McDonald [108], the procedure of seed priming involves soaking the seeds in a suitable solution, and then drying them again. This initiates the germination process, except for radicle emergence [108]. Seed priming promotes the best circumstances for the inoculation and colonization of bacteria in the seed [109]. On the other hand, biopriming is the process of soaking the seeds in a bacterial suspension for a certain amount of time to enable bacterial imbibition into the seed [110]. According to Reddy [111], in order to protect seeds from diseases, biopriming—applying a beneficial bacterial inoculum and keeping the seeds hydrated—is a commonly employed biocontrol technique. The method of soaking seeds in bacterial solution initiates physiological processes within the seed, which prevents the development of plumules and radicles [112], until they are warmed and oxygenated after sowing. PGPR (Plant Growth-Promoting Rhizobacteria) can grow in the spermosphere and multiply in the seed [113] even before sowing. The focus is on seed biopriming since it allows endophytic bacteria to enter through natural pores and prevent adverse stress conditions. Better plant development and strong, even germination may be encouraged by biopriming therapy [114]. Biopriming with rhizospheric bacteria has been reported in crops such as carrot [115], sweet corn [106,116], and tomato [117,118]. Priming has been shown to improve plant growth and production and to be beneficial in the case of biological agents’ efficacy and survival [106,116,118,119]. Since biopriming aids the seed’s ability to survive by offering a healthy environment, seed priming with PGPR results in improved seedling establishment and germination [112]. When it comes to more effective choices and uniformity in germination, biopriming is the most preferred strategy. Although biopriming has been used and described in a variety of ways by many researchers over the years [114,116,120,121,122,123,124] it is still an ambiguous technique that requires further investigation and discussion, particularly in field-level trials.

#### 3.2.3. Seed Encapsulation Technology

Since inks were first microencapsulated in the 1950s, it has been widely recognized that encapsulation techniques may be used to formulate a variety of active substances [125], including extremely creative capsule systems for use in medicine [126,127,128,129,130,131], among others. Publications pertaining to encapsulation have multiplied dramatically over time. Almost 2500 publications were made annually in recent years, of which over 1500 were patents [132]. As such, a wide range of encapsulation techniques are available for the synthesis of biological control agents, particularly microbial agents.

The process of creating a shield or capsule around the active components or cells, such as a microbial or macrobial cells or tissues, is known as encapsulation. According to McLoughlin [133], by creating a definite, stable, and protected microenvironment, microbeads shield inner cells from mechanical, biotic, and abiotic stress as well as unfavorable external environmental conditions. This allows cells to survive and maintain metabolic activity for long periods of time, with the release of cells under controlled conditions once they have adapted to their new surroundings [134]. Because microencapsulation protects bacteria from harmful environmental elements, including pH changes and toxic agents produced during processes, it greatly increases the vitality of microorganisms. Short-chain fatty acids, hydrogen peroxide, bacteriophages, carbonyl-aromatic chemicals, and drying may all be avoided by the bacterial cells with the help of microencapsulation [135]. Immobilized cells have been observed to express alcohol tolerance [136], primarily due to increased changes in the cell membrane [137]. Several fatty acid impurities are present in alginate encapsulation, and these impurities most likely cause the immobilized cells’ fatty acid pattern to change when compared with free cells [138]. A similar effect of tolerance against phenol in *E. coli* was reported [139], which was linked to the cellular membrane’s absorption of the saturated fatty acids found in commercial alginates. Later, Keweloh et al. [139] demonstrated that other components of alginate, in addition to fatty acids, also have a physiological impact on the cell membrane [138]. Another less complex but regulated procedure is the release of microbes from microbeads. There are several ways to crack a micro-envelope, including pressure, diffusion, solvent action, heat, and diffusion [140]. As a result, encapsulated microbes are far more effective than the traditional liquid and powder formulations, with both traditional and sophisticated formulations having advantages and disadvantages of their own [141].

#### 3.2.4. Soil Application

When biocontrol agents (BCAs) are too vulnerable to desiccation, which is typically the case during drought and hot weather, soil treatment is recommended [142]. BCAs control disease by establishing a strong population in the soil. In these situations, niche exclusion also comes into play because the increased quantity of newly introduced microorganisms prevents soil pathogens and other less advantageous microflora from accessing vital nutrients [143]. Both beneficial and pathogenic microorganisms are stored in the soil, and the addition of microbial inoculants to the soil will boost the dynamics of the enhanced bacterial antagonist population and prevent the spread of pathogenic microbes to infection sites. Utilizing cellulosic transporters and binders as well as contemporary thin-film coating methods, several species of *Trichoderma* have also been extensively synthesized and introduced into the rhizosphere regions of seedlings to protect them from soil-borne diseases, including *Pythium ultimum* and *Rhizoctonia solani*. Nonetheless, the primary constraint of fungi as seed coverings persists; hence, they are less able to infiltrate the rhizosphere than bacterial agents [142]. Several attempts have been made to introduce natural substrates colonized by pathogen antagonists into the soil in order to manage a number of soil-borne diseases [144]. Despite the use of aqueous solutions of bioagent propagules for soil inoculation, the distribution of bioagents in the soil may not be uniform. Bankole and Adebanjo [145] concluded that the use of *Trichoderma viride* soil inoculation significantly reduced the level of infection in cowpea seeds infected with *Colletotrichum truncatum* (brown blotch). Applying *Trichoderma harzianum* to the soil effectively controls groundnut stem rot, which is caused by *Sclerotium rolfsii* [146]. Chrysanthemum wilt has been inhibited by an aqueous drench containing conidia of *T. harzianum*, which prevented *Fusarium oxysporum* from re-invading. Weststeijn [147] discovered that inoculating the soil with *Pseudomonas* suspensions at a concentration of 10^8^ cells per gram of dry soil before planting bulbs decreased root rot in tulip trees caused by *Pythium ultimum*. It was discovered that applying *Pseudomonas cepacia* strain N24 to seedbeds at a rate of 500 mL per m^2^ in a greenhouse reduced wilt disease in sunflowers [148].

#### 3.2.5. Foliar Spray Application

Improving spraying conditions while taking the available equipment into consideration is necessary for the effective and efficient application of microbials [149,150,151,152] and the respective formulations with biological control agents. Spraying equipment manufacturers, including Lechler (Metzingen, Germany), Spraying Systems Co./TeeJet Technologies (Glendale Heights, IL, USA), and Delevan Spray Technologies (Widnes, UK), recommend spraying nozzles tailored to different types of crops, growth stages, insecticides, fungicides, and herbicides. However, they often do not specify recommendations for microbial agents and biopesticides. Due to their affordability and efficiency, microbial inoculants are often sprayed using currently available chemical spraying equipment in organic farming and integrated pest management (IPM) strategies. This makes it possible to draw insights from earlier comparative research on chemical pesticides. For example, Dorr et al. [152] tested a variety of hydraulic nozzles and spray mixtures and showed that nozzle design, operating parameters, and spray formulations affect spray characteristics such as droplet size and velocity, liquid density, fan angle, and air inclusion. Moreover, changes in droplet size may affect coverage, while adjustments to fan angle may affect the uniformity of foliar spray. The characteristics of spray droplets can affect the dangers and efficacy of pesticides [153].

*Azospirillum Brasiliense*, *Pseudomonas fluorescens*, and/or *Bacillus* sp. leaf-spraying has also been shown to promote plant development in grasses, including grain crops [154] and pastures, as well as legume grain crops [155,156]. However, Fukami et al. [157] observed and recovered a few *A. brasiliense* cells in corn leaves following leaf spray inoculation, providing a preliminary indication that the effects may be attributed to bacterial metabolites rather than live cells. This was validated in tests using leaf spray of cell-free metabolites. As a result, the effects of leaf spray involving both *A. brasiliense* and *P. fluorescens*, living cells or their metabolites, could be attributed to systemic signaling from shoots to roots, which contributes to root growth and induces mechanisms of resistance to abiotic and biotic stresses [158,159,160]. Tejada et al. [161] discovered that foliar-spraying maize crops with PGPB derived from sewage sludges boosted the protein content of harvested maize seeds by over 30% throughout both studied growth seasons. Furthermore, the protein content of maize seeds was improved by 19% by using PGPB generated from olive oil byproducts [162]. The protein content in soybean seeds was modestly increased after foliar spraying with a high concentration of synthetic PGPB [163].

#### 3.2.6. Root Dip Method

It is well known that pathogens can occasionally be soil- or seed-borne, and they can form host–parasite interactions by penetrating via the root. Therefore, it is crucial to protect the rhizosphere region by using PGPR before colonization in order to have the opportunity to stop the development of a host–parasite interaction. Before being transplanted, seedling roots can be treated with an antagonist spore or cell suspension by dipping them in a microbial inoculant suspension or soaking the biological agents in a nursery bed. In most cases, this technique works well for rice and vegetable crops where transplanting is performed [164]. An experiment by Jambhulkar and Sharma [165] highlights the decrease in bacterial leaf blight of rice that was observed when paddy seedlings were soaked for two hours prior to transplantation in a suspension of *Pseudomonas fluorescens* based on talc. Similar to this, in another experiment conducted in Tamil Nadu, India, dipping rice seedlings in a talc-based formulation of *P. fluorescens* (PfALR1) before transplanting decreased the severity of sheath blight and boosted yield [166]. Nandakumar et al. [167] reported that encouraging results were obtained in controlling the occurrence of sheath blight disease by dipping rice seedlings in bundles in water containing a talc-based formulation of strain mixtures (20 g/L) for two hours, and then transplanting them to the main field.

## 4. Development of Microbial Waste Compound Formulations

Agro-industrial waste can be difficult and time-consuming to compost using traditional methods. The composting process is influenced by a plethora of variables, including raw materials, timeframes, ambient conditions, and more. Prior research demonstrated that the addition of inoculum had no influence on the rate at which organic matter degraded during the composting of wheat straw, agricultural waste, and grape pulp [168,169]. Due to competition between the inoculant and native bacteria, as well as factors such as timing and type, inoculations may not always work well. Thus, researchers have experimented with a number of tactics to enhance the composting process through the use of inoculums. Adding microbial inoculum at various phases of the composting process is a potential strategy. The microbial inoculum can be applied in one, two, or multiple stages of the composting process. The addition of the inoculum at different phases of the composting process has a noticeable effect on physicochemical parameters [170,171]. For example, a study by Zeng et al. [172] showed that as compared to inoculation during the first phase of the agricultural waste (rice straw + bran + vegetable) composting process, the addition of *Phanerochaete chrysosporium* during the second phase promotes a considerable shift in compost maturity. Bacterial inoculation at different stages of rice straw [173], maize straw [174], and citrus peel [175] composting significantly enhanced lignocellulose degradation, thus reducing the C/N ratio and composting period. A review by Fan et al. [176] demonstrated that when lignocellulosic waste was composted, microbial inoculation had an almost 100% favorable impact on temperature, enzyme activity, microbial population, the C/N ratio, and humification, as well as a more than 50% favorable impact on the breakdown of organic matter, GI, N, P, and K. Nevertheless, their research showed that adding microbes to municipal solid waste composting is less successful. This is due to the easily degradable organic matter found in municipal solid waste that can be broken down by native or existing bacteria.

Table 3 provides an overview of the effects of microbial inoculation on the composting of different agro-industrial wastes. Prior research revealed that improved mineralization resulted from introducing bacteria into the composting process [177], accelerated the composting process of oil palm empty fruit bunch (OPEFB) from 64 days to 50 days [178], and enhanced the compost maturity of rice straw and cattle manure by increasing total nitrogen, phosphorus, and potassium content [179]. Due to the enhancement of key enzymes (cellulase, xylanase) and core microbial metabolisms, the re-inoculation of microbial agents *Aeromonas caviae* sp. SD3, *Shinella* sp. XM2, *Rhizobium* sp. S8, *Corynebacterium pseudotuberculosis* sp. SD1, and *Streptomyces clavuligerus* sp. XM, which were screened from rice straw compost, into the composting pile accelerated the degradation of organic matter and coarse fiber content by 7.58% and 8.82%, respectively [180]. Furthermore, in comparison to the uninoculated control treatment, the inoculation of a microbial inoculum comprising *Ralstonia* sp. (LT703298), *Penicillium* sp. (LT703297), *P. aurantiogriseum* (LT703295), and *Acremonium alternatum* (LT703296) enhanced the enzymatic activities of cellulase (15.0 to 19.8%), urease (2.3 to 71.4%), and polyphenol oxidase (0.3 to 28.4%). This resulted in a shorter composting period and an improved maturation rate in the composting of apple tree branches and pig manure [181]. Henry et al. [182] found that adding effective microorganisms (EM) to the composting of pine debris, rice bran, and chicken manure increased the number of thermophiles and, as a result, increased the pace of composting compared to control composting. The findings by Wang et al. [175], demonstrated that adding a bacterial consortium inoculant to the composting of citrus peel, bran, and lime reduced the C/N, organic matter, and moisture while promoting the enrichment of the *Bacillus*, *Sphingobacterium*, and *Saccharomonospora* genera. This improved the degradation of pectin and cellulose. Additionally, adding phosphate-solubilizing bacteria to sugarcane waste composting increased bacterial growth, primarily of the Lactobacillales order. The increased temperature at the beginning of the composting process encouraged the breakdown of the lignocellulosic content, which, in turn, enriched the phosphorus content at the end of the composting process [183,184]. Several researchers have provided findings to help develop microbial waste compound formulations. In Table 3 summarizes microbial waste compound degradation and the most important compounds that are produced.

## 5. Challenges and Limitations in Microbial Formulation Technology

Limitations and challenges in microbial formulation technology utilizing the potential of beneficial soil microorganisms to produce biofertilizers that improve the yield of plants have gained interest in recent years. Although this strategy has seen great success, it is not without difficulties and limitations. One of the main obstacles is the difficulty in reproducing their beneficial effects on plants in the field under constantly changing environmental conditions. Additionally, agricultural communities need to be made more aware of the ecological significance of these microbial formulations as well as the scientific methodologies for using them in the field. To promote their acceptance and effective implementation, outreach and education initiatives are essential. There may also be ethical issues, especially if genetically engineered microbes or non-native species are used in these compositions. The adoption of such techniques may be significantly influenced by society’s acceptance of them. Furthermore, currently existing natural soil microbe populations may provide significant obstacles to the effective application of these inoculants. It is not certain that microbial biofertilizers will perform consistently in a variety of crop varieties and conditions. It might be difficult to choose the most efficient microbial strains for a given agricultural environment. Furthermore, other parameters, including soil type, temperature, pH, and moisture content, might affect how effective these strains are. The short shelf life of microbial preparations is another drawback. The microbes in these formulations may become less viable with time, which could decrease their efficacy in the field. Strict quality control is necessary throughout production to preserve the uniformity and efficacy of these products. Studies on commercial biofertilizers have shown problems with contamination and the presence of unintended bacterial strains. For example, Herrmann and Lesueur [207] analyzed 65 commercial biofertilizers and found that only 37% of these products fulfilled the requirements to be labelled as “pure”. On the other hand, a noteworthy 63% of the biofertilizers that were evaluated showed signs of contamination from one or more bacterial species. In addition, it was discovered that 40% of the tested items included impurities and lacked the designated strains completely. Other limitations include a lack of appropriate carriers for these formulations, poor storage facilities to avoid infection, and unpredictable efficacy owing to severe weather. The absence of important labeling information, such as expiration dates and the identity of the microorganisms utilized in manufacturing, might also cast doubt on the validity of using biofertilizers. The selectivity in the processes of most biofertilizers limits their compatibility with specific chemical pesticides or fertilizers, potentially impacting integrated pest control or nutrient management programs. It is essential for scientists, agricultural practitioners, and policymakers to work together and conduct ongoing research and development to overcome these obstacles and limitations. To promote sustainable farming practices, it is imperative to investigate and capitalize on the possible advantages of microbial formulations while actively resolving their disadvantages.

## 6. Future Aspects

Microbial technology evolved because of fast breakthroughs in science, technology, and development. Technological advancements in this discipline have increased the economic effect of research initiatives. Even though bioremediation has emerged as an effective treatment option for valorizing waste compounds, several hurdles limit its widespread commercial adoption. To completely gain the benefits of bioremediation processes, the following challenges must be addressed.

Many researchers have proven the use of bioremediation to remove poisons from actual waste. It is necessary to extensively explore bioremediation applications to assess their potential for deployment.Most studies focused on batch-scale bioremediation techniques for pollutant removal. The commercial potential of bioremediation as a cost-effective and fulfilling option should be investigated.A multidisciplinary approach is necessary to solve contemporary issues and broaden the practical applications of microbial formulation techniques.Microbial genome engineering can lead to the development of modified microorganisms with improved biodegradation capabilities.Degradation mechanisms, operational factors, and favorable environments for bacteria must all be properly assessed.

Currently, there is a need to pay special attention to the utilization of microorganisms in various ways for the valorization of waste compounds to sustain or restore the native microbiome of the soil bulk or rhizosphere. Reusing raw resources that are currently discarded as garbage is one of the main tenets of the Circular Economy Package, which was enacted in December 2015. In the future, farmers’ product options may be expanded by fertilizer preparations made by microbial solubilization with sustainable raw materials. This appears to be a reaction to the current trend of looking for a new generation of agricultural goods produced using ecologically friendly practices and bio-based raw materials. Increasing the recycling of products and their reuse brings measurable benefits for both the environment and the economy. The actions promoted by EU regulations are aimed at ensuring the better use of raw materials, products, and waste. Furthermore, encouraging the broader use of recycled nutrients would facilitate a more resource-efficient use of these nutrients and aid in the development of the circular economy. Research on potential sustainable methods should focus on substituting waste inputs for mineral fertilizers and providing plants with a special type of root-associated microbe that can depolymerize and mineralize nutrients attached to organic matter. Because a variety of industrial, municipal, and agricultural operations generate enormous volumes of nutrient-rich “waste” that is currently disposed of but may one day be converted and used as fertilizers, waste inputs can be obtained more sustainably than mineral fertilizers. Another reason is that nutrients attached to organic matter are less likely to leak or volatilize because they are more stable in the soil than fertilizers made of minerals. It is advisable that in the future, efforts are made to learn how to use and comprehend these microbes and their functions, as well as how to efficiently colonize plants using soil inoculation techniques.

## 7. Conclusions

The majority of current research on complex compound-degrading microbial consortia has focused on native consortia isolated from the environment. However, we predict that future investigation will focus on artificial microbial consortia. The development of microbial consortia systems using metabolic engineering and synthetic biology demonstrates high degradation potential, offering a new technique for the effective exploitation of complicated substrates and the restoration of the environment. The effective introduction of microorganisms to the soil system ensures their sufficient colonization, tailored to carry out specific functions and to gain insight into the mechanisms of how soil microbes boost plant growth under conditions when nutrients are not readily available. Co-composting with microbes has the added benefit of improving degradability and reducing nutrient valorization in compost. Although the incorporation of microbial cultures may improve composting efficiency, the economic feasibility of microbial culture costs remains an essential concern in future research.

## Figures and Tables

**Figure 1 microorganisms-12-00541-f001:**
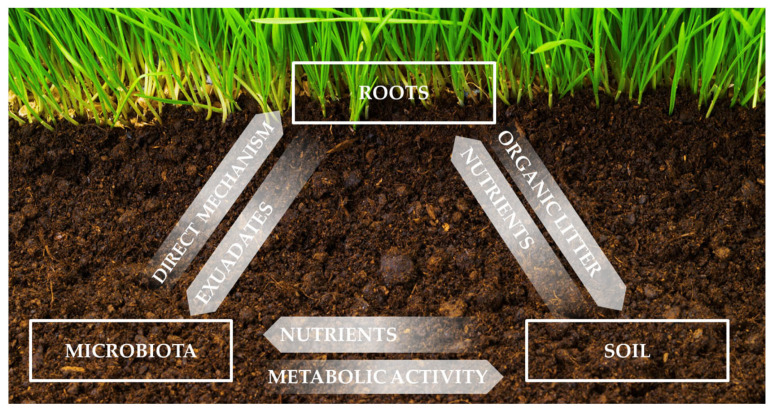
Graphical representation of below-ground components of the soil system.

**Figure 2 microorganisms-12-00541-f002:**
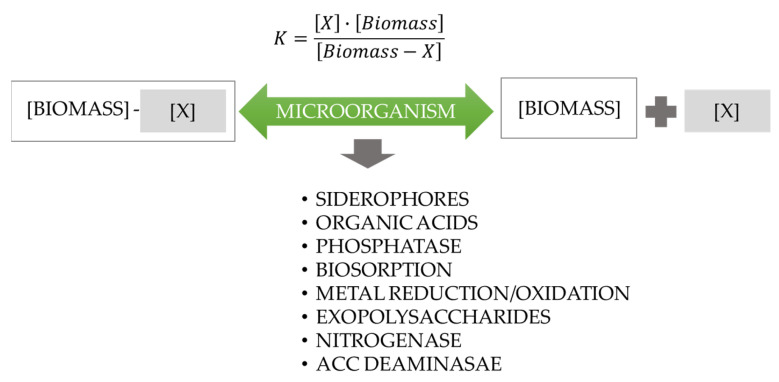
Model illustrating the direct and in direct mechanism of microbiota action in the soil system. The equation represents K = constant; X = nutrient of interest.

**Figure 3 microorganisms-12-00541-f003:**
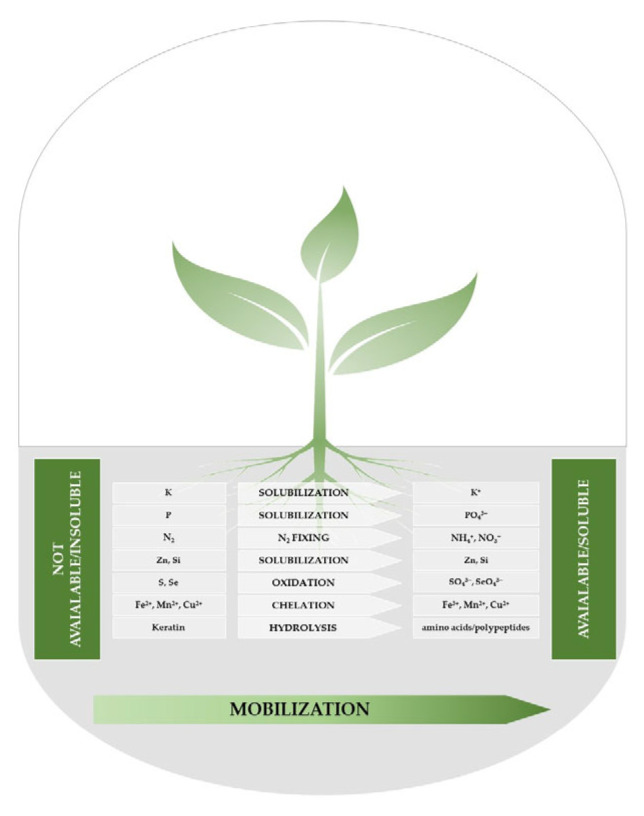
Model illustrating nutrient liberation processes from not available to available forms to be utilized by the plants.

**Table 1 microorganisms-12-00541-t001:** The characteristics of waste biomass.

Waste Biomass	Possible Source of Nutrients	References
Ash	Phosphorous, Potassium, and Zinc	[16]
Biochar	Nitrogen, Phosphorous, Potassium, and Zinc	[17]
Bones meat meal	Phosphorous, Potassium, Zinc, Iron, and Selenium	[18]
Spent coffee grounds	Potassium, Magnesium, Selenium, Phosphorus, and Iron	[19]
Blood meal	Nitrogen, Phosphorous, Potassium, Zinc, Iron, and Selenium	[20]
Feathers meal	Amino acids and Nitrogen	[21]
Spent mushroom substrate	Nitrogen, Phosphorous, and Humic acid	[22]

**Table 2 microorganisms-12-00541-t002:** Application of synthetic microbial consortia in the degradation of various waste compounds (adapted from [95]).

Wastes Compound	Composition of Consortia	Performance	Reference
Cattle manure	*Methanosarcina acetivorans* and *Methanosaeta thermophila*	Biogas production increased by 45%	[96]
Chicken manure	Nitrogen-converting bacteria	Reduced ammonia loss by 59%	[97]
Biogas residue	Bacteria, fungi, actinomycetes, and yeasts	Drying contribution accounted for 79%	[98]
Food waste	*Bacillus amyloliquefaciens* B59, *Bacillus licheniformis* B58, *Bacillus haynesii* A31, and *Bacillus amyloliquefaciens* B11	The volatile solids removal improved by 10%	[99]
Municipal waste	Six plastic-degrading bacterial	Improved degradation of different plastics	[100]
Mustard biomass	*Saccharomyces cerevisiae* and *Fusarium incarnatum*	Bioethanol production increased 33 mg/mL	[101]
Poultry manure	Six strains including *Bacillus subtilis* and *Streptomyces rutgersensis*	The concentration of odorants reduced 58–73%	[102]

**Table 3 microorganisms-12-00541-t003:** Microbial degradation of waste compounds. (Adapted from [184]).

Waste Compounds	Microbial Treatment	Culture Concentration	Composting Conditions	Impact on the Entire Composting Process	References
Mushroom residue	*Paenibacillus* GX 5 *Paenibacillus* GX 7 *Paenibacillus* GX 13 *Brevibacillus*, *GX* 5 *Brevibacillus*, *GX* 7 *Brevibacillus*, *GX* 13	2 mL 100 g^−1^	C/N-12, T–(57 °C), MC–(60–24%), pH-(8)	Increased microbial contact, extended thermophilic period, and improved rate of lignocellulose and organic matter decomposition.	[185]
Mushroom residue and wood chips	*Aspergillus*, *Penicillium* *Bacillus*, *Streptomyces*	0.2% (w/w^−1^)	C/N–(22), T- (58.4 °C), MC-(50%), pH-(7.8)	extended thermophilic stage, improved cellulose and hemicellulose breakdown efficiency, and optimal microbial community structure.	[186]
Chicken manure and maize straw	*B. licheniformis*, *B. amyloliquefaciens*, *Ureibacillus thermosphaericus*, *B. megaterium*, *Geobacillus pallidus*, *B. pumilus*, *Geobacillus* sp. *Paracoccus denitrificans*	200 mL of 1 × 10^8^ CFU mL^−1^	C/N-(21), T-(68.4 °C), MC-(55.6–42%), pH-(8.7)	Increased germination index, NO_3_ content, prolonged thermophilic stage, reduced volatile solids contents, improved humification and compost maturity level.	[187]
Chicken manure and rice husk	*Ureibacillus terrenus* BE8 and *B. tequilensis* BG7	5% (v/w^−1^)	Total C (263 g kg^−1^), and Total N (34 g kg^−1^), T-(65 °C), MC-(78.1%)	Improved germination index values, faster compost maturity through early stimulation of many important microorganisms, and superior phytotoxicity-free compost compared to the control treatment.	[188]
Pig manure and wheat straw	Microbial agent solution consisting of photosynthetic bacteria, actinomycetes, yeasts, and lactic acid bacteria	40 mL 10 kg^−1^	Total C (41.2 ± 0.5%), Total N (1.79 ± 0.03%), T-(68.4 °C), MC-(55%)	The possible hosts of ARGs have changed because of changes in ARG profiles and bacterial populations, which has increased the removal of ARGs in their entirety.	[189]
Rice straw	Compound bacterial agent screened from rice straw composts: *Aeromonas caviae* sp. SD3 (KR868995.1), *Shinella* sp. XM2 (CP015736.1), *Rhizobium* sp. S8 (KF261556.1), *Corynebacterium pseudotuberculosis* sp. SD1 (CP020356.1) and *S. clavuligerus* sp. XM (CP032052.1)	1% (w/w^−1^) of 1 × 10^9^ CFU mL^−1^ cell concentration	C/N-(30), MC-(65%)	Improved the degradation of organic matter and coarse fiber content by 7.58% and, 8.82% due to the enhancement of core microbial metabolism.	[180]
Chicken manure, rice bran and pine waste	Bacteria: *Bacillus* spp., *Alicyclobacillus* spp., *Pseudomonas* spp., *Lactobacillus* spp., *Pediococcuss* spp., and *Actinomycetes*. Fungi: *Rhizomucor pusillus*, *Aspergillus* spp.	0.2% (w/w^−1^)	C/N-(28.4), T-(65 °C), MC-(60 to 40%), pH-(8.5)	Enhanced mineralization, composting rate, and microbial population and variety.	[182]
Rice straw biogas residue and rice straw	*A. niger* CICIMF0410 and *P. chrysosporium* AF 96007	1% (v/w^−1^) of 1 × 10^8^ CFU mL^−1^ cell concentration	C/N-(32), T-(68.3 °C), MC-(60%)	Reduced the time required for decomposition of organic matter, removed the toxicity risk for crops and promoted the stability of the compost.	[190]
Swine manure and spent mushroom substrate	Microbial suspension of lignocellulose-degrading microorganism’s consortium consisting of *Bacillus*, *Brevibacillus*, *Paenibacillus* and *Lysinibacillus* genera	10% (v/w^−1^)	Mixture ratio (1:1), T-(68 °C), MC-(60%), pH-(7.6)	Promoted the changes of the bacterial community in the mesophilic phase and reduced the risk of ARGs in the final compost.	[191]
Maize straw and canola residue	*Phanerochaete chrysosporium*	1 × 10^8^ CFU mL^−1^	C/N-(25), T-(60 °C), MC-(52%), pH-(8.17)	Improved lignin degradation during the cooling stage, enhanced compost humification.	[192]
River sediment, rice straw, vegetables, and bran	*Phanerochaete chrysosporium*	0.5% (v/w^−1^)	C/N-(30), T-(69 °C), MC-(60%), pH-(8.6)	Enhanced the passivation of copper and reduced the effect of pH on the bioavailability of heavy metals.	[192]
Dairy manure and sugarcane leaves	Thermophilic lignocellulolytic microbes screened from dairy and sugarcane leaves compost samples: *B. licheniformis* (TA65), *A. nidulans* (GXU-1) and *A. oryzae* (GXU-11)	2% (w/w^−1^)	C/N-(30), T-(55 °C),	Enhance the lignocellulose degradation process and the humification process, as well as the mineralization of organic carbon.	[193]
Pig manure and corn stalk	Compound bacterium agent comprised of *Acinetobacter pittii*, *B. subtilis* sub sp. *Stercoris* and *B. altitudinis*	1% (v/w^−1^) of 1 × 10^9^ CFU mL^−1^ cell concentration	C/N-(30), T-(67.3 °C), MC-(60%), pH-(8.8)	Increased the number of biomarkers, prolonged the thermophilic stage, reduced the amount of human disease-related functional genes, and improved fertility and longevity.	[194]
Citrus peel. bran and lime	The bacterial consortium which was screened from citrus peel compost samples	3% (w/w^−1^)	C/N-(25), T-(65 °C), MC-(60%), pH-(8.5)	Decreased C/N, organic matter, moisture, pectin and cellulose content, and enhanced the richness and diversity of the bacterial community.	[175]
Cattle manure and wheat stalks	*B. subtilis*	0.5% (w/w^−1^)	C/N-(25), MC-(60%), pH-(7.61)	Promoted changes in ARGs and removed many pathogenic bacteria.	[195]
Wheat straw, rice, corn and soybean	Actinomycetes: *Streptomyces* sp. H1 (KX641927.1), *Mycobacerium* sp. G1 (KY910181.1), *Micromonospora* sp. G7 (LC333394.1) and *Saccha-romonospora* sp. T9 (NR074713.2)	3 mL kg^−1^ of 1 × 10^9^ CFU mL^−1^ cell concentration	C/N-(30), T-(63 °C), MC-(50 to 60%), pH (9.4)	Improved 34.3% lignocellulose degradation and 8.3% enzyme activity.	[196]
Pig manure and apple tree branches	Microbial inoculum: *Ralstoinia* sp., *Penicillium* sp., *Penicillium aurantiogriseum*, and *Acremonium alternatum*	2% (v/w^−1^)	C/N-(30), T-(77 °C), MC-(60%), pH-(8.1)	Enhanced cellulase, urease, and polyphenol oxidase activities and promoted the succession of the bacterial community structure.	[181]
Corn straw and dairy manure	Thermo-tolerant actinomycetes *Streptomyces* sp. H1, *Streptomyces* sp. G1, *Streptomyces* sp. G2 and Actinobacteria bacterium T9	2% (v/w^−1^) of 1 × 10^9^ CFU mL^−1^ cell concentration	C/N-(30), T-(57 °C), MC-(60%)	Enhanced cellulase activities and increased degradation of cellulose, humic substances content.	[197]
Food waste and maize straw	Cold adapted microbial consortium comprised of stains *P. fragi* (KY283110), *P. simiae* (KY283111), *Clostridium vincentii* (KY283112), *P. jessenii* (KY283113) and *Iodobacter fluviatilis* (KY283114).	1% (v/w^−1^) of 1 × 10^8^ CFU mL^−1^ cell concentration	C/N-(18), T-(45 °C), MC-(66%)	Improved the breakdown of organic materials at low temperatures and encouraged a shift in the succession and composition of the bacterial population.	[198]
Dairy manure and rice straw	Psychrotrophic-thermophilic complex microbial agent (PTCMA): *B. diminuta* CB1, *Flavobacterium glaciei* CB23, *A. niger* CF5 and *Penicillium commune* CF8	10 mL kg^−1^ of 1 × 10^8^ CFU mL^−1^ cell concentration	C/N-(32), T-(63 to 45 °C), MC-(60%), pH-(8.2 to 8.4)	In colder areas, raising the temperature of the composting pile, greatly enhancing the compost’s maturity, and proposing PTCMA injection are all useful strategies.	[199]
Sugarcane industry waste	Phosphate-solubilizing bacteria: *P. aeruginosa*, *Bacillus* sp., *Lactobacillales*, *Bacillales*, *Pseudomonas* sp., *Clostridiales*	8 L mg^−1^ of 1 × 10^8^ CFU mL^−1^ cell concentration	C/N-(30), T-(60 °C)	Elevated bacterial development, mostly of the Lactobacillales order, which results in the heaps heating up in the first stage of composting and having an increased phosphorus content at the end.	[183]
Rice straw, soil, vegetables, and bran	*Phanerochaete chrysosporium*	2% (v/w^−1^) of 1 × 10^6^ CFU mL^−1^ cell concentration	C/N-(30), T-(58 °C), MC-(60%), pH-(8)	reduced the lead’s toxicity and enhanced the composting bacterial community’s diversity	[200]
Chicken manure and rice straw	Ammonia-oxidizing bacteria	5% (v/w^−1^) of 1 × 10^6^ CFU mL^−1^ cell concentration	C/N-(25), T-(57 °C), MC-(60 to 70%), pH-(7.4)	Reduced nitrogen loss and ammonia emissions by the conversion of ammonium to nitrite and improved bacterial community abundance.	[174]
Rice straw	Cellulase producing bacteria: *B.* *licheniformis* 1-1v and *B. sonorensis* 7-1v	1% (v/w^−1^) of 3.6 and 6.8 × 10^7^ CFU mL^−1^ cell concentration	C/N-(35.8), T-(54 °C), MC-(35%), pH-(8.1)	Lowered the composting period by 40 to 43%, which improved the quality of the compost and led to a greater drop in the total organic carbon and C/N ratio.	[179]
Vegetable waste: cattle manure: sawdust	*Phanerochaetechrysosporium* (MTCC 787)	10^7^ to 10^8^ spores g^−1^ of compost	Compost mixture ratio (5:4), T-(64 °C), MC-(65%), pH-(7.5)	improved the volatile solids reduction over the uninoculated compost treatment by 1.45 times in trial 2 (the initial phase) and 1.7 times in trial 3 (the thermophilic phase).	[201]
Rice straw and goat manure	EM: lactic acid bacteria, yeast and phototrophic bacteria.	5% (v/w^−1^)	C/N-(32.4)	Improved the mineralization in composting process.	[177]
Wheat straw and cattle manure	Ammonium-oxidizing bacteria: Bacillaceae (strain T-AOB-2, M-AOB-4 and MT-AOB, 2–4)	5% (v/w^−1^) of 1 × 10^8^ CFU mL^−1^ cell concentration	C/N-(30), MC-(65%)	Enhance bacterial activity and encourage the production of humic compounds by lowering total and dissolved organic carbon.	[202]
Chicken manure, furfural residues and bagasse	Exogenous microbes (VT) and indigenous microbes (M3T)	0.5% (v/w^−1^)	C/N-(30), T-(50 to 58 °C), MC-(55%)	Increased urease, protease, and cellulase activity, as well as a faster rate of temperature increase.	[203]
Maize straw and pig manure	*B. subtilis*, *B. licheniformis*, *Phanerochaetechrysosporium*, *Trichoderma koningii*, *Saccharomyces cerevisiae*	0.1% (w/w^−1^)	C/N-(27.7), T-(66 °C), MC-(60%)	Improved rate of temperature increase, increased micronutrients (N, P, K), enhanced decomposition of organic carbon, improved germination index.	[204]
Wheat straw and dairy manure	Microbial agent: *A. niger*, *Saccharomyces cerevisiae*, *Lactobacillus plantarum*, *Lactobacillus acidophilus*, *B. megaterium*, *S. albogriseus* and *B. subtilis*	0.2% (v/w^−1^)	C/N-(16), T-(60 °C), MC-(60%), pH-(8.0)	Raised essential bacterial network interaction, reduced possible pathogen abundance, and increased composting maturity and overall organic carbon decomposition.	[30]
Rice straw and cattle manure	*Malbranchea cinnamonmea*, *Gloephyllumtrabeum*	10 mL kg^−1^	C/N-(25), T-(73 °C), MC-(65%), pH-(8.5)	Strengthened nutrients and humus carbon, enhanced lignocellulosic fungal variety and relative abundance, and promoted decomposition of cellulose, hemicellulose, and lignin.	[205]
Rice straw and swine manure	*Kitasatospora phosalacinea* C1, *Paenibacillus glycanilyticus* X1, *B. licheniformis* S3, *Brevibacillus agri* E4 and *Phanerochaete chrysosporium*	Not mentioned	C/N-(27.5), T-(62 °C)	Increased degree of maturity and improved pace of temperature increase.	[206]
Wheat straw and swine manure	*Gloephyllum trabeum*	1 × 10^8^ spores kg^−1^	C/N-(27), T-(73 °C), MC-(60%)	Shorten maturation period, increased decomposition rate of cellulose, hemicellulose and lignin, influencing fungal community by increasing relative abundance of *Aspergillus*, *Mycothemus* and *melanocapus*.	[205]

Note: MC-moisture content; C/N-carbon nitrogen ratio; CFU-colony forming unit; T-temperature.

## Data Availability

No new data were created or analyzed in this study. Data sharing is not applicable to this article.

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
