# Peer review of "Production of a Rich Fertilizer Base for Plants from Waste Organic Residues by Microbial Formulation Technology"

_microorganisms, 2024, doi:10.3390/microorganisms12030541_

Round 1
Reviewer 1 Report
Comments and Suggestions for Authors
This study focuses on different methods of sustainably introducing nutrients from agriculture food waste into the soil by microbial formulation technology. The topic is interesting. However, there are several issues needed to be addressed.
1 Ln 105 - Ln 113, the relevant reference should be provided.
2 Table 1, it is weak to show the typical possible sources. The details and relevant references should be added in this table.
3 Ln 366, Microbial consortia part, important details and logic should be described. Please make the key points stronger than current version. You can add more findings and relevant references.
4 Page 17, the table content is difficult to follow the information. You should modify the table style of this page.
5 Future aspects, it should be significantly improved. It should include the scientific advances and limitations of the microbial formulation technology to date. Also, please provide more essential key points on future aspects.
Comments on the Quality of English LanguageModerate english editing is required.
Author Response
We (Authors) thank you for your review and valuable suggestions & corrections pointed. We have agreed with your comments and each point raised were critically evaluated and modified accordingly. The corrections were highlighted with red color font in the Manuscript. Please find the attached file for the point by point response to your comments.

Reviewer 2 Report
Comments and Suggestions for Authors
Dear authors,
This work is very interesting, well organized and well written, however there are some comments should be considered before publication such as:
- The abstract seems like addressing the aims of the review, therefore the abstract should be rewritten to include the most important points such as the aims, the approaches or methodology, the findings, and the contribution to the field
- In figure two, the Eq. is not indicative because it is not known what does it refer to, as well to write Eq. you should write its reference
- Please magnify the figure in order to show the bottom part clearly
- In line 180, change mg year not mg/year
- The conclusion should be extended to include the most important findings and the contribution to the field. As well, "The current review aims" is not suitable in the conclusion, it is more suitable in the abstract or introduction
- Please update the references from 2020 - 2024
Author Response

(The authors gave the same response as above.)

Reviewer 3 Report
Comments and Suggestions for Authors
The suggestions for
“Formulation of waste compounds into nutrient-rich source for plants by microbial formulation technology”
Suggested title: “Production of a rich fertilizer base for plants from waste organic residues by microbial technology”
Line 13
Delete: “…soil…”
Lines 31,32,33,34
Delete: “The equilibrium of the soil and rhizosphere's microbiota could be significantly impacted, so nutrient rich secondary raw materials can, nevertheless, offer a valuable foundation for modern mineral fertilizers—if the new formulations avoid overloading the soil-plant nexus with potentially toxic substances”.
Replace with: “The equilibrium of the soil and rhizosphere's microbiota could be significantly affected, and nutrient rich secondary raw materials can offer an improved mineral nutrition, if the new formulations avoid overloading the soil-plant nexus with potentially toxic substances”.
Lines 45, 46
delete blank spaces in the text.
Line 111
Delete: “…up. And, …”
Replace with: “…up, and…”
Figure 2: not very understandable figure, caption of the figure to be completed.
Line 58
Delete: “enclosed in the matrix of secondary nutrient flows”
Line 173, 174
At least one bibliographic citation is required.
Line 180
Delete: “…mg year…”
Replace with: “…Mg * year -1 ” or “…Mg / year …”
Line 256
Delete: “.”
Line 334
Delete: “…B…”
Replace with: “…Bacillus…”
Line 356
Add: “…Arbuscular Mycorrhiza…” after (AM)
Line 416
Add: at the beginning “Biopriming is the seeds hydration utilizing a biological or a biocontrol ingredient.”
Line 418,419
Delete: “…The procedure of seed priming involves soaking the seeds in any solution that contains the necessary priming ingredient, then drying them again.…”
Replace with: “…the procedure of seed priming involves soaking the seeds in a suitable solution, then drying them again.…”
Line 421, 422
Delete: “..Biopriming is the phrase for hydration utilizing any biological ingredient, 421
or in this case, a biocontrol agent..”
Line 425
Delete: “…In…”
Replace with: “…in…”
Line 430
Add: (Plant Growth-Promoting Rizhobacteria) after PGPR
Line 438
Delete: “…and offer…”
Replace with: “…offering…”
Line 439, 440
Delete: “…stand establishing…”
Line 454
Delete: “…By…”
Replace with: “…by…”
Line 463
Delete: “…The immobilized…”
Replace with: “…the immobilized …”
Line 470
At the end of the sentence only a full stop
Line 494, 496
Delete: “…T.…”
Replace with: “…Trichoderma …”
Line 497
Delete: “…S.…”
Replace with: “…Sclerotium …”
Line 498
Delete: “…T.…F…”
Replace with: “…Trichoderma …Fusarium….”
Line 501
Delete: “…P.…P…”
Replace with: “…Pthium …Pseudomonas….”
Line 537
Delete: “…20 g/l…”
Replace with: “…20 g/L….”
Line 566
Add: “…(Oil Palm Empty Fruit Bunch)…” before “OPEFB”
Line 579
Delete: “…effective microorganism…”
Replace with: “…Effective Microorganisms….”
Line 621
Add: “may” before “decreases”
Line 624, 625, 626, 627
Delete: “…The current review aims to explain the effective introduction of microorganisms to the soil system that ensures its sufficient colonization, tailored to carry out specific functions and to gain insight into the mechanism and understanding of how soil microbes boost plant growth under conditions when nutrients are delivered in not available, waste-born form.…”
Replace with: “…The current review aims to explain the introduction of microorganisms effects to the soil system to ensures its sufficient colonization to carry out specific functions, and to improve understanding of how soil microbes boost plant growth under conditions when nutrients are delivered in not available, waste-born form.….”
Line 640
Add: “Structural variability” before “and Differentiation of Niches in the Rhizosphere…”
Line 751
Add: “Metabolic” before “load”
Line 761
Delete: “ Bacillus Firmus”
Replace with: “Bacillus firmus”
Author Response

(The authors gave the same response as above.)

Reviewer 4 Report
Comments and Suggestions for Authors
It is a review paper inform about formulation of waste compounds into nutrient-rich source for plants by microbial formulation technology. Utilization of waste into fertilizer as a part of closing the loop of nutrients carries a risk of biological imbalance of the soil microbiota. The Authors present waste biomass as a source of nutrient material. The current review aims to explain the effective introduction of microorganisms to the soil system that ensures its sufficient colonization, tailored to carry out specific functions and to gain insight into the mechanism and understanding of how soil microbes boost plant growth under conditions when nutrients are delivered in not available, waste-born form. Therefore, the paper fits the journal scope. The overview of different forms of microbial formulations are interesting for the readership of the journal.
Regarding the quality, the paper is rather well written although some weak points were found and below are my specific comments for the Authors.
The Authors can give examples of foliar spray application.
At the Figure 2. Model illustrating the direct & in direct mechanism of microbiota action in the soil system
can use word enzymes (phosphatases, nitrogenase, dehydrogenases etc.)
translate what is K and [X] in formula in page 4.
At the Figure 3. Model illustrating nutrient liberation processes from not available to available forms to be 149
utilized by the plants.
-the letters are too small. The Authors should minimalist the figure of plant and give bigger the font the information of mobilization processes at the roots space.
Correct some editorial mistakes
page 5 line 190 small letter at the beginner of sentence
page 13 line 359 problem with bracket.
Author Response

(The authors gave the same response as above.)
